# Implementing Smart Sustainable Cities in Saudi Arabia: A Framework for Citizens' Participation towards SAUDI VISION 2030

**Abood Khaled Alamoudi** [1,2,*] , **Rotimi Boluwatife Abidoye** [2] and **Terence Y. M. Lam** [3]

1 Department of Architecture, College of Architecture and Planning, Imam Abdulrahman bin Faisal University, Dammam 31451, Saudi Arabia
2 School of Built Environment, University of New South Wales, Kensington, Sydney, NSW 2052, Australia
3 Department of Architecture and Built Environment, Northumbria University, Newcastle NE1 8ST, UK; terence.lam@northumbria.ac.uk
* Correspondence: a.alamoudi@unsw.edu.au

**Abstract:** Cities in Saudi Arabia need to expand rapidly due to the rapidly growing urban population. To develop smart sustainable cities (SSC), human, social, and environmental capital investments must be expanded beyond just focusing on technology. There have been several cities that have adopted smart city labels as recognition of the advantages of smart cities. Many countries acknowledge the value of citizens' involvement in public urban planning and decision making, but it is difficult to evaluate their impact and compare it to other factors. This study aims to develop a citizens' participation framework, identify any additional stakeholder's management measures (SMM) (in addition to the ones previously developed by the authors), and explain the relationship with citizens' participation level (CPL) for driving SSC. Three rounds of the Delphi method were conducted to structure and validate the framework by the decision maker in the field of urban planning and reach a consensus of understanding the drivers of SSC. The study group was limited to 25 participants because this study focuses on the perspective of decision makers toward CP. Mean score (MS) ranking and Kendall Coefficient were used to confirm the importance of these additional stakeholders' management measures. The results suggest three main component structures of the conceptual framework, which are SMM, CPL, and Citizens' Participation Recruitment (CPR), which are all necessary for smart sustainable city outcomes (SSCO) for achieving the Future Sustainable Cities Plan (FSCP) within the context of Vision 2030 and government policy in Saudi Arabia. Using the proposed framework will enable all the stakeholders to gain a deeper understanding of SSC and their complex natures from a conceptual and practical standpoint. The contribution to knowledge of this study is by developing a conceptual framework that can support the implementation of SSC, and by providing an understanding the CPR standards and the involvement of citizens in urban development, which eliminates any debate regarding SSC.

**Keywords:** smart sustainable cities; citizens' participation framework; citizens' participation recruitment; citizens' participation level; stakeholder's management measures; Saudi Arabia

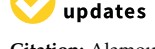



## 1. Introduction

Smart sustainable cities (SSC) have been acknowledged globally in response to rapid urbanisation and enormous consumption which require attention and collaboration from diverse professions [1]. The United Nations [2] estimated that 90 per cent of the increased population would live in cities by 2050. Globally, urban areas consume about 70 per cent of the world's natural resources, which significantly contribute to greenhouse gas emissions [3]. Saudi Arabia has also witnessed an expanding in urban development, wherein the cities expand rapidly. As a result, several consequences of urban issues have occurred, such as traffic congestion, insufficient transportation facilities, resource depletion

at an accelerated pace, and a certain issue that is not being addressed culturally [4]. Due to the lack of comprehensive planning frameworks, rapid population growth coupled with weak institutional structures has led to sprawling and lopsided development [5].

Gassmann, Böhm, and Palmié [6] and Ramaprasad, Sanchez Ortiz, and Syn [7] argue that advanced technology such as Artificial Intelligent (AI) and Internet of Things (IOT) play an important role in promoting SSC. Moreover, opportunities such as innovation, entrepreneurship, and job creation can be attracted by SSC, which leads to economic growth and flourish, as well as equitable and inclusive urban environment. However, Alamoudi, Abidoye, and Lam [8], Bibri [9], and Embarak [10] debate that cities that are driven by technology are most likely to exacerbate social inequalities, whereas cities that consider human involvement and participation in the SSC development are more likely to do so equitably, justly, and particpatorily. Human needs and interests are prioritised and promoted not only to privileged groups, but also marginalised communities. Ultimately, cities have been challenged to balance between utilising Information and Commination Technology (ICT) and ensuring that citizens' participation (CP) is applied.

In response to these rapid urbanisation challenges, many sustainability measurement models and frameworks have developed, including but not limited to the Leadership in Energy and Environmental Design (LEED), the Building Research Establishment Environmental Assessment Method (BREEAM), and the Comprehensive Assessment System for Built Environment Efficiency (CASBEE) [11]. Over the past few decades, several governments have adopted a variety of urban sustainability systems. Even though most of these systems were designed for buildings, the advantages of ICT have not been taken into consideration [9]. The complexity and sophistication of urban areas are incredible, and it requires a holistic framework such as the SSC concept to manage its complexity [12,13]. Smart cities have been viewed by various scholars as a new paradigm created to accommodate rapid urbanisation and economic development [10,12,14], while other spectators believe the smart cities concept is not new [1,15]. The phrase, smart cities concept, was introduced in 2007/2008 when ICT was introduced to businesses, institutions, and the public; on the other hand, the concept of smart growth originated in the late 1900s during the growth movement [1,12]. Based on academic expertise, smart cities can be viewed as a complex ecosystem that is supported by technological infrastructure and means of transforming the engagement, participation, and learning of citizens [16]. Although scholars acknowledge that SSC is primarily a techno-centric concept because it leverages ICT, they claim that social, cultural, economic, and environmental aspects are crucial [17–20].

SSC are those that have extensively integrated ICT with their urban systems [21]. In the past, urban governments have struggled to achieve sustainability goals through ICT [22]. ICT technology and community participation are absent from policies and strategies for sustainable urban development. By using a smart sustainability framework, urban development initiatives can be driven by technology and CP with positive environmental, economic and social outcomes [23]. Sustainable urban planning is considered one of the key outcomes of smart cities, which are open to embracing new technologies. Smart city initiatives, however, do not provide evidence of how sustainability outcomes are achieved [24]. It is possible to collect and analyse datasets for urban intelligence functions using ICT approaches. A framework can be used to formulate decision-making strategies to achieve SSC [3].

Urban planning and challenges can be addressed with ICT [16]. However, transitioning to SSC requires an understanding of urban governance [25]. In urban governance, there are two types: (1) traditional government that is characterised by centralisation, little or no public involvement, and private partnerships, namely, "traditional government", and (2) one that emphasises decentralisation, public participation, partnerships, and consensus building, referred to as "modern governance" [26]. Urban planning and governance are also interconnected and involve a variety of stakeholders [27]. Almughairy [28] suggests that planning, governance, and implementation must be coordinated for regional development to be successful. It is imperative that urban planning and its governance are linked in order

for urban sustainability to be achieved [27]. According to Almughairy [28], by utilizing a region's uniqueness in planning and implementation, any community and its residents can be assured of a prosperous future. Other scholars such as Al-Hathloul [29] argue that Saudi Arabia's management system has a national and local focus and is centralised at the national level.

Understanding the six characteristics of smart cities is essential; these include: smart people, smart economy, smart mobility, smart living, smart governance, and smart environments [14]. Various technologies are used by SSC in order to achieve a sustainable lifestyle and steady, healthy quality of life (QoL). Sustainable development is primarily achieved by creating, deploying, and promoting ICT. As a result, it is referred to as a "new TechnoUrban phenomenon" [3]. It has been widely discussed that CP plays a significant role, but there has not been evaluation of its effects to compare its contribution to other influences [30–32]. Several studies have found that countries that empower CP perform significantly better in urban projects than countries that do not [33–35]. For example, in 2016 the Future Saudi Cities Program (FSCP) was established by the Minister of Municipal and Rural Affairs (MoMRA) in collaboration with UN-Habitat to capture the urban challenges in Saudi Arabia [36]. FSCP aims to promote spatial balance, reduce urban sprawl, and develop a decentralised planning framework in Saudi Arabia for a sustainable city. QoL, environmental protection, and economic competitiveness are all part of FSCP's business objectives. Boosting the productivity, equitability, social and ecological well-being of Saudi cities is the objective of FSCP. Smart Sustainable Cities Outcomes (SSCO) are seen in three-primary areas as business objectives: 'QoL', 'economic competitiveness', and 'environmental protection' [37]. QoL as a measure of social sustainability aims to have productive and prosperous cities through high-quality urban design that is equitable and socially inclusive, and which has an adequate and efficient infrastructure. Environmental protection considers reducing sprawl, enhancing spatial balance development, and promoting the environment. Economic competitiveness is distinguished by producing better financing, a greater degree of well-being, and better employment opportunities [38]. These outcomes rely heavily on ICT. An example is using sensors to collect air and water quality data, waste management data, and other environmental information. In addition, cities can boost their economic competitiveness by leveraging ICT to attract investment, promote entrepreneurship, create jobs, and grow their economies. Moreover, social inclusion, community engagement, cultural diversity, and public safety can all be enhanced through ICT in cities, as well as crime and violence reduction through ICT. UN-Habitat's objectives for SSCO leave CP untouched, creating gaps between ICT and CP as facilitators of urban sustainability and citizens engaging in the development of cities as stakeholders. To achieve the implementation goal, it is necessary to study and address these gaps. Therefore, there is a need to develop a comprehensive framework that can work together to bridge the gap between ICT and CP to achieve SSC.

Recently, the 12 Vision Realization Program of strategic importance for the government of Saudi Arabia was established and named VISION2030. Its objective is to improve the lifestyle of individuals by creating an ecosystem that boosts participation in cultural, entertainment, and sports activities, develop events for communities that enhance liveability, and enhance the ranking of Saudi cities [38]. According to UN-Habitat [37], the Saudi Government's aim is to make Saudi's cities considered one of the top 100 cities globally by 2030. Some Saudi cities may be able to improve their social, economic, and environmental conditions through this program [17].

There is a need to mitigate the undesirable consequences of rapid urbanisation by adapting the SSC framework. This study aims to develop a citizens' participation framework, identify any additional stakeholders' management measures as revealed by the stakeholders involved in the FSCP program, and explain the relationship with CPL for driving SSCO for FSCP and VISION2030. In particular, we raise the research question: How can the citizens' participation be enhanced to achieve the business objectives of FSCP, i.e., SSCO? Three rounds of the Delphi method were conducted to fulfill the study's aim. The paper contributes significantly to the literature on SSC by developing a framework that supports CP in the form of two-way communication with all urban development stakeholders. In other words, local governments and other stakeholders can better understand how smart cities operate through a practical framework.

The novelty of the proposed framework lies on the component of the framework. It contributes to understanding the issue conceptually and practically. The novelty of the framework introduces a new approach to archive and understand the involvement of CP in the development of SSC. In addition, its components, including SMM, CPL, SSCO, and Citizens' Participation Recruitment CPR, differentiate it from previous frameworks. Another advantage is the ability to identify the most appropriate SSC implementation measures for different cities, as well as the expectations of stakeholders, to assist the decision makers. Meaning, it aims to support the communication between the government and decision makers and identify what is expected from the citizen and vice versa to achieve SSCO. This novel framework contributes to advancing the field of urban planning and provide significant understanding for policymakers, practitioners, and researchers.

The structure of this study is as follows. Firstly, we present the current research problem. Subsequently, a literature review of various frameworks that support the SSC concept is presented. The research methodology is illustrated in the Section 3, and research questions are discussed in detail; in the Section 4, the results, discussions, and validation of the framework are presented. As a conclusion to the paper, the Section 5 discusses the study findings, implications, and limitations, and future studies will be proposed.

## 2. Literature Review

### 2.1. Review of Sustainability Frameworks

Previous studies show that many governments around the world have been implementing different approaches towards smart cities and sustainable urbanisation [3,24,39]. Globally, over 30 rating systems have been used to measure and monitor sustainability in environmental, economic, and social aspects [40]. Measurements (see Table 1), such as the Urban Management Program of UN-Habitat [41], Melbourne City Council's City plan [42], the government of Singapore's Green Plan [43], have been used to guide and monitor urban sustainability. Smartainability aims to include most of the urban sustainability indicators and allows the estimation of how smart cities promote sustainability [13]. The first introduced rating system was BREEAM, which was developed in 1990 [44]. Out of these rating systems, three are the most accredited and famous systems: LEED, the Estidama pearl rating system, and the Global Sustainability Assessment System (GSAS) [45]. However, GSAS is recognised to function in countries in the Middle East and North Africa (MENA) region, including the Gulf and Saudi Arabia [40]. Because of the range of explanations of SC and urban sustainability, many frameworks and ranking systems have been developed (see Table 1 for the sustainability frameworks) [22]. There are few studies that compared the impact of these frameworks and their related indicators [22]. However, there is very little or nothing in the literature about weighting these frameworks and their indicators in order to adopt the suitable one, nor about the integration of ICT with sustainable cities.

**Table 1.** Best Practices Frameworks for Urban Sustainability.

| Name of the Framework | Description | Number of | | Limitation |
| | | Category * | Indicators * | |
|---|---|---|---|---|
| ISO 37120 * | Sustainable development and resilience are assessed holistically through an integrated and holistic approach to quality of life QoL and service delivery in cities [22] | 17 | 100 | Some of the indicators were eliminated due to space limitations [46] |
| RFSC * | The European Cities Toolkit is a free tool aimed at promoting and improving the integrated urban development actions of cities and urban territories [22] | 4 | 24 | It supports only local European Union authorities to restricted access |
| BREEAM | The purpose of an assessment method is to improve, measure, and certify the sustainability of large-scale development plans in terms of social, environmental, and economic factors [22] | 9 | 62 | Difficulties in controlling the quality assurance and high cost to obtain it [47] |
| LEED-ND for Neighbourhood Development | Using standards to distinguish whether the neighbourhood is environmentally improved; green certification is applied to the neighbourhood context [22] | 5 | 53 | The cost of earning such credits is high, while few points are earned for meeting their criteria [48] |
| CASBEE for Urban Development (CASBEE-UD) | Assessing the effects of a conglomeration of buildings on the environment at the urban scale [22] | 6 | 76 | Incorporates some of the issues in the main categories into the management side, instead of the main category of sustainable urbanisation itself [49] |
| STATUS * | Developing locally relevant tools to help establishing targets for urban sustainability through a joint initiative by researchers and practitioners [22] | 8 | 46 | It supports only local European Union authorities to restricted access |
| SustainLane | Ranking system of 50 of the country's largest cities to recognise the depths, challenges, and potential of each major city's management policies [22] | 16 | 46 | A description of how weights were assigned to individual initiatives or why certain initiatives were included in the city rankings is not provided [50] |
| UN-Habitat CPI | An extensive set of indicators that measure progress toward the Habitat Agenda and the Millennium Development Goals includes 20 key indicators, 8 checklists, and 16 extensive indicators [22] | 5 | 42 | The definition of prosperity does not address all kinds of urban typologies such as slums [51] |
| UN-Habitat SDG * | Analyses how countries are performing on SDGs on an average. High SDG rankings are strongly related to high natural resource demands per person [52] | 17 | 169 | Resource security is not well represented among the goals and targets, so a more complete and carefully constructed SDG will not have a significant impact on results [52] |

* Category: the impact of indicators, * Indicators: A measure that captures information about a complex phenomenon, * ISO 37120: An indicator of the QoL and sustainable development of cities, * RFSC: An acronym for the Reference Framework for Sustainable Cities in Europe, * STATUS: Sustainability Tools and Targets for the Urban Thematic, * SDG: for Sustainable Development Goals, which was developed by UN-Habitat.

## 2.2. Review of Smart Cities Frameworks

Table 1 summarises the urban sustainability frameworks. Many researchers see urban sustainability from three pillars/dimensions, which are economic, environmental, and social [53,54], while Khogali [55] added a fourth pillar, which is the cultural pillar. Every

pillar/dimension comprises measures based on a set of indicators and sub-indicators. On one hand, Table 2 shows the frameworks and indicators that are used to measure SSC in many countries. For example, the United States of America uses LEED, France uses RFSC, Japan uses CASBEE, and England uses BREEAM. On the other hand, Table 2 shows a wide range of frameworks designed for measuring smart cities, where each framework uses a number of categories and indicators.

According to Alamoudi, Abidoye, and Lam [14], SSC falls under six major domains. Each domain is measured using a set of indicators and sub-indicators. These domains are Smart Economics, Smart Environment, Smart Governance, Smart People, Smart Living, and Smart Mobility. Alamoudi, Abidoye, and Lam [56] assessed SSCO using three sets of measures. First: primary areas of urban sustainability proposed by FSCP. Second: urban indicators utilised by FSCP. Lastly: the most common indicators and sub-indicators revealed by literature review for measuring urban sustainability.

Alamoudi, Abidoye, and Lam [56] suggested the following indicators and sub-indicators for SSC, which are Smart Economics: innovative spirit, entrepreneurship, economic image, trademark, flexibility of labour market, and E-business; Smart Environment: attractivity of natural conditions, pollution, environmental protection, sustainable innovation, and safe transport systems; Smart Governance: participation in decision making, public and social services, transparent governance, and E-government; Smart People, level of qualification, inclination to lifelong learning, social and ethnic plurality, flexibility creativity, cosmopolitanism/open mindedness, and participation in public life; Smart Living: cultural facilities, health conditions, individual safety, housing quality, education facilities, touristic attractivity, and social cohesion; Smart Mobility: local accessibility, international accessibility availability of ICT-infrastructure, sustainable, and innovative and safe transport systems.

**Table 2.** Selection of Best Practice Frameworks for Smart Cities.

| Name of the Framework | Description | Number of Category * | Number of Indicators * | Limitation |
|---|---|---|---|---|
| European Smart Cities Ranking | Ranking of European cities developed by the University of Technology Vienna [22] | 6 | 64 | It requires open data in order to function the best [57] |
| The Smart Cities Wheel | By examining all key components that make a smart city, Boyd Cohen developed an integrated framework to support them [22] | 6 | 26 | Especially in developing nations, limit the concept to smaller and emerging cities [57] |
| Smart city benchmarking in China | Developed as part of a Chinese project and used for evaluating 28 Chinese cities' smartness [22] | 5 | 43 | The model was built based on a comparison with other cities' strategies, planes [58] |
| Triple-helix network model for smart cities performance | For measuring the performance of smart cities, a model links the interrelationship between their components [22] | 5 | 45 | Its main focuses are on digital services only [59] |
| Smart City PROFILES | Five SC indicators, with a focus on energy efficiency and climate change [22] | 5 | 21 | It focuses on climate change and energy [60] |
| City Protocol | Creating city-centric approaches that benefit citizens is the goal of an international collaborative innovation that starts in Amsterdam [22] | 9 | 190 | This program ended in 2018 although all the insightful information is still accessible [61] |
| CITYkeys | Providing a holistic measurement framework (under the EU H2020 program) [22] | 20 | 73 | The data set and indicators are calculated based on the availability and reliability of the needed data [62] |

* Category: the impact of indicators, * Indicators: A measure that captures information about a complex phenomenon.

There is a need to better understand how smartness and sustainability are related and interconnected [63]. As shown in Tables 1 and 2, a significant number of contradictions in theory and practice are rooted in the technological world, yet the definition of the SC concept is not unified [22,24,64–66]. As a result, its definition, character, and dimensions are shaped by the scholar's background and how it is applied [24,67]. Although the SSC concept has only recently been introduced to academic discourse as a means to promote sustainable urban development, it still remains an area of nascent empirical research [21].

As proposed by Alamoudi, Abidoye, and Lam [68], a strong smart sustainable city system relies on four factors: knowledge, awareness, citizen participation, and opinion about government policy. On the other hand, the relationship between stakeholders' management measure (SSM) and CPL has been tested and validated by Alamoudi, Abidoye, and Lam [8]. As suggested, Regulation, Collaboration, Legitimates, and Control are the most important stakeholder management variables that drive CPL. In additional, the impact of CPL on the SSCO has also been tested and validated by Alamoudi, Abidoye and Lam [56], which demonstrates the following CPL variables: Accountability and Responsibly Transparency, Participation, and Inclusion. This paper defines an additional SSM as revealed by the stakeholders involved in the FSCP program, which will be examined and determined by utilizing the Delphi method.

Figure 1 shows the relationship between CPL, SSM, CPR, and SSCO. This research hypothesises that E-government/ICT, engaging/empowerment, and socio-cultures factors are significant measures in the stakeholder management process and can influence CPL and SSCO.

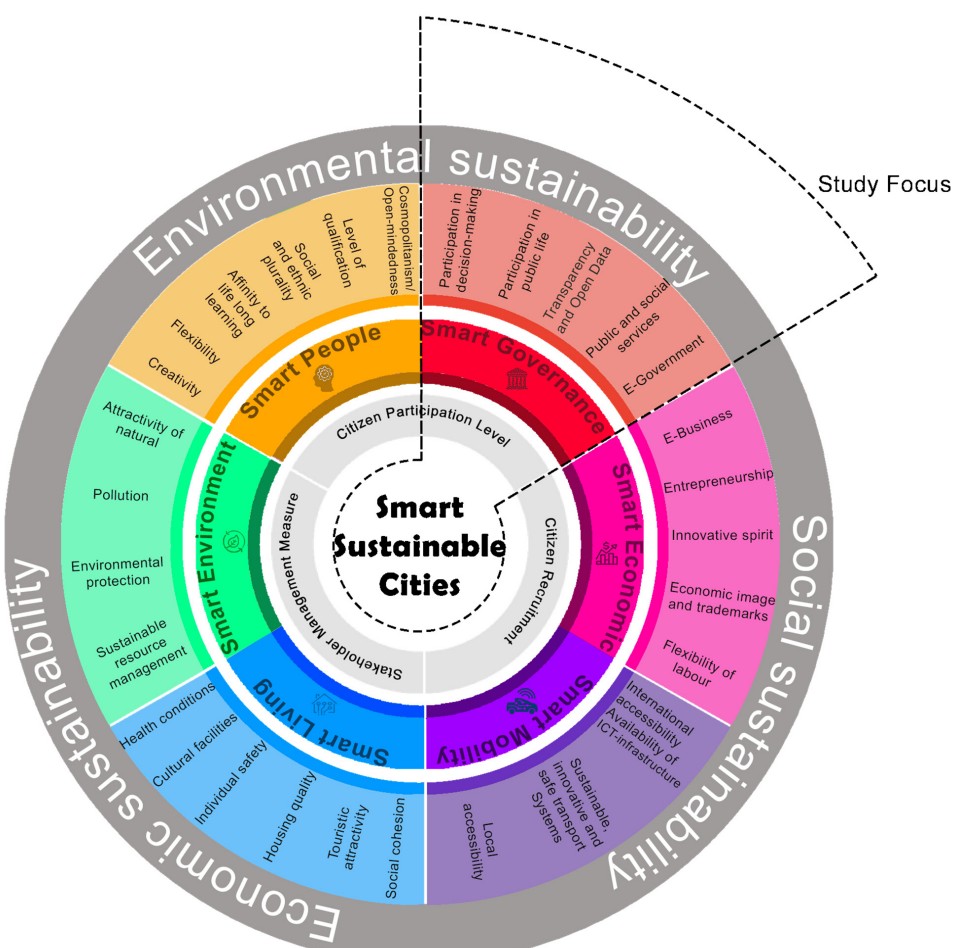

**Figure 1.** A framework for holistic Smart Sustainable Cities and the areas of study focus. Adapted from: Alamoudi, Abidoye, and Lam [56].

## 3. Methodology

To assess SSCO from adopting citizens' participation measures and stakeholder communication enhancement measures, and, hence, validate the stakeholder management framework for achieving smart sustainable cities outcomes, a Delphi method was adopted. It contributes to understanding the SSC by structuring and validating results from experts on built environments [69,70]. Three rounds of questionnaires were sent to experts for brainstorming, narrowing down, and ranking the most important components of the framework [71]. The first round aimed to express experts' opinions to confirm the framework and identify any additional measures and relationships for the citizens' participation framework for driving SSCO for FSCP, which will support better communication between the stakeholders involved in smart cities development. In the second round, the questionnaire was sent out with feedback from the first round for validation of the responses of the structure of the framework by comparing it with the consolidation of others' opinion. In the third round, the responses from the second round were presented to the participants to gain a consensus experts' opinion. Moreover, the participants were asked an open-ended question to confirm and identify the influencing measures and to explain the relationships between them. The first survey was structured into five sections: (1) general questions about the participants, (2) foundation of the Citizens' Participation framework (Recruitment), (3) foundation of the framework—E-government/ICT Factors, (4) foundation of the framework—Engaging/Empowerment Factors, and (5) foundation of the framework—Social-Cultural factors. The items were scored on a five-point Likert scale: "Least Important (1)", "Fairly Important", "Important", "Very Important", and "Extremely Important (5)", respectively [72]. The data were collected from a mix of participants groups, including government representatives who work at FSCP and/or (MoMRA), professionals, and academics. The participants were reached through the webpage of their companies or via their LinkedIn profiles, which contained their position, experience, and involvements. The research team made initial contact with potential participants by directly sending a recruitment invitation email. If no response was received, a friendly reminder was sent out to the participants. If no response was received after this reminder, this indicates no interest to participate, and the potential participant was then removed from the survey.

### 3.1. Data Collection for Delphi Method

To obtain a comprehensive view of stakeholders, an online questionnaire was distributed [73]. By mixing local practitioners and academics in Saudi Arabia, any misunderstanding, lack of knowledge, and lack of observational evidence are eliminated [74]. This study, therefore, collected opinions from stakeholders in the FSCP sector, from professionals (urban planners, architectural designers, real estate developers), as well as representatives from governments (FSCP officers from MOMRA). In addition to policymakers and UN-Habitat, academics also participated. To determine the importance of each performance predictor and outcome, a five-point Likert scale was used [75].

Table 3 shows that majority—24 (96.0%)—of the respondents are male, while only 1 (4.0%) is female. This revealed that males participated more than females in this study. According to the World Bank [76], it is no surprise that female response rates are low in Saudi Arabia as most urban professions are dominated by men. The Saudi Arabian government and the World Bank reported that the current total female workforce in Saudi Arbaia is 20.4%, and in the Built Environment field the percentage drops to 12%. This study captures 17.4% of women, which validates the work [76]. Nevertheless, the homogenous group and the study group size are limitations of this study, and it forms a separate area for further research. Moreover, 9 (36.0%) of the respondents are in the age group of 40–49, 8 (32.0%) are in the age group of 30–39, and the age group of 50 and above has a total number of 8 (32.0%) participants. As regards geographical area, more than half—14 (56.0%)—of the respondents are from the Riyadh region, 9 (36.0%) are from the Eastern region, while only 2 (8.0%) are from the Makkah region. The geographical area was limited to these three major cities due to the fact that all the decision makers are located in one of these cities.

Furthermore, as regards field of profession, 11 (44.0%) respondents specialise in urban planning, 7 (28.0%) specialise in Information Technology (IT), while only 4 (16.0%) are professionals in Management. More than half, i.e., 21 (84.0%), of the respondents practice in the public and private sector, while only 8 (16.0%) practice in the private sector and in academia. Four of the participants hold PhDs, which indicates that they are well educated and can contribute to the achievement of the goal of this current study. Capturing the perspective of experts who have good experience is essential. About 10 (40.0%) of the respondent indicated that they possess 16–20 years of experience in their field of practice, while 7 (28.0%) possess 10–15 years of experience, and 8 (32.0%) hold 21 years and more of experience. In addition, 6 (24.0%) of the respondents are CEOs, 5 (20.0%) are managers, and 4 (16.0%) are academics.

**Table 3.** Descriptive statistics of respondent demographics.

| Characteristics | | Frequency (N-25) | Percentage |
|---|---|---|---|
| Gender | Male | 24 | 96.0 |
| | Female | 1 | 4.0 |
| | Others | 0 | 0 |
| Age | 30–39 | 8 | 32.0 |
| | 40–49 | 9 | 36.0 |
| | 50 and above | 8 | 32.0 |
| Region | Riyadh Region | 14 | 56.0 |
| | Makkah Region | 2 | 8.0 |
| | Eastern Region | 9 | 36.0 |
| Field of profession | Architecture | 3 | 12.0 |
| | Urban Planning | 11 | 44.0 |
| | Management | 4 | 16.0 |
| | IT | 7 | 28.0 |
| Sector of practice | Public Sector | 17 | 68.0 |
| | Private Sector | 4 | 16.0 |
| | Academia | 4 | 16.0 |
| Years of experience | 10–15 | 7 | 28.0 |
| | 16–20 | 10 | 40.0 |
| | 21-more | 8 | 32.0 |
| Position in your firm | Partner/Founder | 1 | 4.0 |
| | Principal | 4 | 16.0 |
| | CEO | 6 | 24.0 |
| | Manager | 5 | 20.0 |
| | Supervisor | 2 | 8.0 |
| | Professor | 2 | 8.0 |
| | Associate Professor | 1 | 4.0 |
| | Assistant Professor | 4 | 16.0 |

### 3.2. Data Analysis Techniques

The relationship between the categorical variables can be determined in several ways. Mean Score (MS) helps to determine the significance of a variable to the others in terms of the most important to the least important. In addition, data were analysed using Regression Analysis. The standardised coefficient represents the strength and direction of the relationship between a predictor variable and the response variable, with a positive coefficient indicating a positive relationship and a negative coefficient indicating a negative relationship [8]. The Statistical Package for the Social Sciences version 26.0 software (SPSS) (Chicago, IL, USA) was utilised, which is widely used in the fields of social science, business, and education.

The significance of the independent and dependent variables was evaluated using the MS technique [77]. Analysis of the collected data was conducted to determine if there

were any additional stakeholder management measures (SMMs) that were associated with them. MS is widely used to evaluate the significance of variables in built environment studies [78–80]. The MS rankings of the variables were calculated using Equation (1) [79]. The data collected were analysed using SPSS software for cross-tabulations, relationships, and groupings.

$$M = \frac{\sum s}{n} \tag{1}$$

where *M* represents the mean score, s is the participants' score based on a Likert scale, and n is the total number of participants.

### 3.3. Data Reliability

In statistics, Cronbach's alpha represents an estimate of the data's reliability or consistency. A Cronbach's alpha value of 0 indicates that there is no consistency among the items in the test, while a value of 1 indicates perfect consistency [81]. In order for a scale to be considered reliable, it must have a Cronbach's alpha coefficient greater than 0.5 [82]. To calculate Cronbach's alpha, the following Equation (2) was adopted.

$$alpha = \frac{k}{(k-1)} * \left(1 - \frac{sum(vi)}{sum(ve)}\right) \tag{2}$$

where *k* is number of items in the test, *vi* is the variance of each item in the test, and *ve* is the variance of the test as a whole.

In terms of the CP and indicators of SSCO, all of the variables show higher reliability. Table 4 shows that the average response values are higher than 5. On average, the respondents were neutral about the following variables: Citizens' Participation Recruitment, E-government/ICT Factors, and Engaging/Empowerment Factors, while the respondents on average fairly achieved Social-Cultural factors.

**Table 4.** Reliability of data.

| Factors | Number of Items | Cronbach's Alpha |
| --- | --- | --- |
| Citizens' Participation Recruitment | 4 | 0.733 |
| E-government/ICT Factors | 14 | 0.724 |
| Engaging/Empowerment Factors | 11 | 0.843 |
| Social-Cultural factors | 17 | 0.664 |

Additionally, considering the background of participants (public, private, and academic), Kendall's Coefficient of Concordance was utilised to determine the degree of consensus between the groups [83]. For the responses that were collected from the first round, all identical responses were removed and consolidated, then factors were grouped into categories to make it easier for panellists to compare when returned for the next round [70]. In terms of responses' validation, during the next round the experts were asked to verify that their responses have been interpreted correctly and to verify and refine the category. The second round narrowed down the factors based on different perspectives and backgrounds. The responses were validated, similar to the previous round. Moreover, the participants were requested to select the most important factors. The final round was the ranking round, where the participants were asked to rank the most important factors until the result reached a consensus. In addition, recommendations of the proposed framework for better communication were examined and validated. Kendall's Coefficient of Concordance ranges between 0 and 1, with values closer to 1 being the strongest indication of agreement and values closer to 0 being the weakest. [72]. Equation (3) was developed by Siegel, Castellan, and Me Graw-Hill [84] to calculate the Kendall's Coefficient.

$$W = 12 \frac{\sum_{i=1}^{n} (Ri - R)2}{P2(n3 - n) - pT} \tag{3}$$

where $n$ = number of factors, $Ri$ = ranks assigned to the $i$ the factor, $R$ = mean value of the $Ri$ values, $P$ = number of respondents, and $T$ = correction factor for the tied ranks.

## 4. Results and Discussion

A qualitative and quantitative study was conducted to examine the framework's development and qualification, involving a number of participants who reported on their understanding of SSCs and different factors affecting its development. The study aims to develop a framework in Saudi Arabia through three rounds of the Delphi method. Data were categorised under different themes according to emerging contexts.

### 4.1. First Round of Delphi Method

Table 5 shows the frequency distribution of variables. The majority—24 (96.0%)—of the respondents, reported that the provision of SC vision is extremely significant, and it has an MS score of 3.8. It is believed that a smart city will decrease costs, improve QoL, and improve the efficiency of services. Some of the key aspects of an SC might include sustainable infrastructure and buildings, advanced transportation systems, energy efficiency, advanced communication systems, robust public safety systems, access to high-quality healthcare, and strong education systems [85]. Moreover, a little more than half—17 (68.0%)—of the respondents reported that random recruitment is extremely important, with a score of MS 3.7, while only 4 (16.0%) reported it as least important. As suggested by Carson and Martin [86], it is an effective way to reduce bias that may result from the participants, and it is a promising technique to promote participation in decision making. As regards decision makers' interaction level, 17 (68.0%) of the respondents reported that decision makers' interaction level is extremely important in smart cities development, and that has an MS score of 2.7, while only 1 (4.0%) reported this as important. Effective community depends heavily on the availability of data, yet maintaining the confidentiality of information depends on the authorised access level of the participant [87]. In addition, majority—23 (92.0%)—reported 'governments to promote monitoring and accountability to its citizens' as extremely important. Lindquist and Huse [88] argue that balance principles of citizen accountability have been debatable and operationalised. However, ICT and digital tools should be leveraged to achieve the balance of accountability of citizen participation. With respect to one-way stakeholders' interaction level, only 8 (32.0%) reported that one-way stakeholders' interaction level is an important factor, while 5 (20.0%) reported that this factor as extremely important, and it has an MS score of 2.2. According to Piqueiras, Canel, and Luoma-aho [89], the ability to communicate is essential to engage citizens in urban development. In addition, shift from one-way communication to two-way communication is essential to respond to the rapid urbanisation. Majority—24 (96.0%)—of the respondents reported that the provision of a smart cities vision is extremely important, and it has an MS of 2.19.

**Table 5.** MS Ranking of Performance Predictors.

| | Level of Adoption | | | | | Mean Score | SD ** |
|---|---|---|---|---|---|---|---|
| | LI * | FI * | I * | VI * | EI * | | |
| Smart cities provide vision | 1 (4.00) | 00 (0.00) | 00 (0.00) | 00 (0.00) | 24 (96.0) | 3.8664 | 0.82927 |
| Random recruitment | 4 (16.0) | 1 (4.0) | 3 (12.0) | 0 (0.00) | 17 (68.0) | 3.7627 | 0.90557 |
| The existing decision makers' interaction level | 00 (0.00) | 00 (0.00)) | 1 (4.00) | 7 (28.0) | 17 (68.0) | 2.7415 | 1.07433 |

**Table 5.** *Cont.*

| | Level of Adoption | | | | | Mean Score | SD ** |
|---|---|---|---|---|---|---|---|
| | LI * | FI * | I * | VI * | EI * | | |
| Citizen-centric E-services | 00 (0.00) | 3 (12.00) | 3 (12.00) | 3 (12.00) | 17 (68.0) | 2.5193 | 1.06707 |
| Governments to promote monitoring and accountability to its citizens | 00 (0.00) | 00 (0.00) | 1 (4.00) | 1 (4.00) | 23 (92.00) | 2.4873 | 1.02098 |
| Development delivery | 00 (0.00) | 00 (0.00) | 1 (4.00) | 1 (4.00) | 23 (92.00) | 2.4492 | 0.9728 |
| One-way stakeholders' interaction level | 00 (0.00) | 5 (20.00) | 8 (32.00) | 7 (28.00) | 5 (20.00) | 2.2032 | 0.8943 |
| Two-way stakeholders' interaction level | 1 (4.00) | 00 (0.00) | 00 (0.00) | 00 (0.00) | 24 (96.0) | 2.1921 | 0.8821 |

* Least Important, Fairly Important, Important, Very Important, Extremely Important, ** Standard Deviation.

### 4.2. Second Round of Delphi Method

Table 6 shows the result from both the round one and round two Delphi study and the consensus reached regarding the associated factors by the experts. These 14 factors were proposed by the expert participants in the Delphi survey. Citizens' trust refers to the level of confidence and belief that members of a community have in their government and its institutions [90]. It is an important factor in the functioning of a healthy democracy, as it helps to ensure that citizens feel that their voices are being heard and that their needs are being addressed by those in positions of power. Citizens' knowledge refers to the understanding and awareness that members of a community have about their rights and responsibilities as citizens, as well as about the issues and challenges facing their community [91]. From the perspective of a citizen living in an SSC, they may experience a number of benefits such as: Increased efficiency, Improved public safety, Better QoL, and Greater access to information [92]. Cultural factors can influence the way that citizens in a city experience and perceive their environment, including their attitudes towards technology and the use of it in their city [93]. Some examples of cultural factors that may affect technology adoption include values and beliefs, social norms and expectations, and level of technological literacy. Citizens' visibility refers to the level of attention or exposure that an individual receives from the government. It is important for individuals to carefully consider the level of visibility [94]. It is important to recognise that everyone's experience is unique, and an individual's marital status does not necessarily define their ability to participate in their community or in public life. In general, all citizens have the right to participate fully in their communities and to have their voices heard, regardless of their marital status [95]. Gender can be a significant factor in an individual's ability to participate in their community [96]. To promote gender equality and inclusivity, it is important for communities and public institutions to recognise and challenge these barriers to participation and to create opportunities for the full and equal participation of all members of the community, regardless of their gender [97]. To summarise, Table 6 shows the result of the Delphi survey for both Round One and Round Two for CPR. The rankings of these factors have similar MS and ranking in both Delphi rounds. However, the citizens' gender has been de-escalated to be less important than citizens' age while the remaining CPR remain in the same sequence.

**Table 6.** Result of Delphi survey Round One and Round Two for citizens recruitment.

| Associated Factors | Round 1 | | Round 2 | |
|---|---|---|---|---|
| | Ranking | Mean Index | Ranking | Mean Index |
| Factors—Citizens' Trust | 1 | 4.92 | 1 | 5 |
| Factors—Citizens' Knowledge | 2 | 4.91 | 2 | 4.96 |
| Factors—Citizens' Perspectives | 3 | 4.9 | 3 | 4.925 |
| Factors—Citizens' Cultural factors | 4 | 4.79 | 4 | 4.89 |
| Factors—Citizens' Relevance | 5 | 4.77 | 5 | 4.87 |
| Factors—Citizens' Visibility/Publicity | 6 | 4.76 | 6 | 4.86 |
| Factors—Citizens' Marital Status | 7 | 4.46 | 7 | 4.56 |
| Factors—Citizens' Employment Status | 8 | 4.14 | 8 | 4.26 |
| Factors—Citizens' Spatial Behaviour | 9 | 3.98 | 9 | 4.08 |
| Factors—Citizens' Ethnicity | 10 | 3.82 | 10 | 3.92 |
| Factors—Citizens' Religion | 11 | 3.79 | 11 | 3.72 |
| Factors—Citizens' Income | 12 | 3.11 | 12 | 3.05 |
| Factors—Citizens' Gender | 13 | 2.93 | 14 | 2.23 |
| Factors—Citizens' Age | 14 | 2.85 | 13 | 2.46 |
| N | | 25 | | 25 |
| Kendall Coefficient | | 0.783 | | 0.574 |
| *p*-value | | 0.000 | | 0.000 |

### 4.3. Third Round of Delphi Method

Common Theme 1: The Engagement, Management, and Adoption of ICT by Stakeholders in Smart City Planning

The result of this round reveals the consensus that the engagement of stakeholders is an imperative process in having the desired smart city of choice. Their involvement in the planning of smart cities will make a whole lot of difference as they will be from different professional backgrounds and levels. A participant opined that stakeholders should be fully engaged as they are important individuals in building an enabling and very active society. Similarly, one of the participants said that "*We can engage stakeholders by three aspects: The first aspect is the organizational aspect*"; "*The second aspect is physical organization. The third aspect is a good ICT infrastructure to achieve the goals of smart cities*". This further validates the position of participants that stakeholders should be empowered by giving them some autonomy to operate within their jurisdiction as far as the building of SSC is concerned. "*I think stakeholders are very important to be engaged in any development and I would like to see them very active in society*". Urban development must be filled with stakeholders from different backgrounds and different levels.

Furthermore, the knowledge and understanding of stakeholders about smart cities hold a significant responsibility in the organisation of time and other related resources for smart city building. A participant said that "*management is essential to measure the participants availability and measure their contribution and measure their inputs and measure their knowledge*". As much as the management of external stakeholders is important, the adoption of ICT leverage cannot also be overemphasised in the building of smart city. The majority of the participants were of the opinion that the ICT system should ensure the process of the type of information to share, when to share it, and how to share the information with the citizens. This will allow an effective and automatic process in communicating the ideas from policymakers down to the citizens. Moreover, communication technology should consider different variables in measuring the goals of stakeholders with respect to the available resources. A participant said that "*an effective and efficient way to collaborate, coordinate and communicate* via *ICT is to define the level of power and to know your participants*". Another respondent suggested that "*my thoughts in designing a system would be to take into consideration the who, what, where, and when to use the information*". The system must include variables that help achieve the goals such as governance of the data, creating the brain of the cities and assigning the task to the right people.

Common Theme 2: The Role of Citizens' Participation in the SSC as a Driver of Sustainable Cities

The SSC is dependent on the cooperation of citizens in almost every section of the development process. The survey results for round three show that the majority of the participants are in agreement that the citizens should participate fully in the urban development that will lead to sustainable cities. It is noteworthy that a participant suggested that the citizens should be empowered for decision making; however, their level of autonomy should be regulated by the relevant stakeholders. A participant said that "*The engagement of citizens participation should not be limited to just get their voices but also to be part of the development*". The findings further reveal that citizens and policymakers must have a cordial relationship that will lead to bridging different forms of gaps that could lead to hold-up in development of some city areas. A participant also shared their thought about the participation of citizens in development of SSC, as they feel that citizens' participation must be properly prioritised in decision making in adopting SSC, in agreement to the respondent who claimed that "*I totally support the engagement of citizens participation, however, to an extent to have only a maximum of the partnership of power with the government*". Further, "*the higher stakeholder's participation the better for urban development which leads to better outcomes*".

Additionally, the participants highlighted some of the important roles of citizens in the development process of SSC. A participant revealed that citizens should have access to government data as well as a countable voice in the government bodies. However, data must be governed and encrypted for security and protection purposes [98]. Similarly, another participant opines that "*the role of citizen participation must be very high in the neighbourhood level while being very low at the national level*".

*4.4. Toward Smart Sustainable Cities Framework*

SSC is regarded to be all-inclusive and is carried out to improve social sustainability, economic sustainability, and environmental sustainability. The three rounds of the Delphi survey focus on obtaining the opinions of experts on SSC with pre-defined questions used to identify and validate their responses. Round three of the survey reveals that citizens and stakeholders play an interrelated and dependent role in the development of SSC. The different categories of stakeholders will need the cooperation of citizens to accomplish the bigger picture of SSC. The majority of participants were of the opinion that citizens should be allowed to make certain decisions since they are mostly affected by the overall decisions of the government stakeholders. Hence, their voices should be acknowledged and considered in every decision process in order to ensure an all-inclusive decision making for the development of SC. The educational and demographic background of stakeholders should be considered before assigning them to project management. This will ensure that they are optimally performing since they have similar backgrounds in city building and project management. Moreover, data security and management should be given priority to maintain the safe identity of all citizens without violation of any sort in driving the smart sustainable cities.

A framework for developing SSC should be based on a comprehensive and participatory planning process that involves all stakeholders, including citizens, government, businesses, and community organisations. This study develops citizen participation for smart sustainable cities framework (CPSSCF) (see Figure 2). It is a combination of research domains and experts' experience used to verify the accuracy and quality of data, models, and systems. The structure of the CPSSCF consists of SSM, CPL, and CPR to effectively identify, engage, and communicate with stakeholders in order to manage their expectations and ensure that their interests are taken into account in decision making. Knowledge of SC and FSCP was examined. A consideration was made of the awareness from the perspective of both the government and the citizens. There is a desire on the part of the CP in this development. As a final point, some opinions were discussed regarding some urban agendas. Furthermore, an essential part of the framework is SMM. SMM consists of four critical factors, which are Regulation, Collaboration, Legitimates, and Control which are

factors that support the increase of CPL [8]. Regulation refers to the act of controlling or directing something according to a set of rules or laws. Collaboration refers to the act of working together with others to achieve a common goal. Legitimates refer to something that is lawful, proper, or in accordance with the rules. Control refers to the ability to direct or manage something.

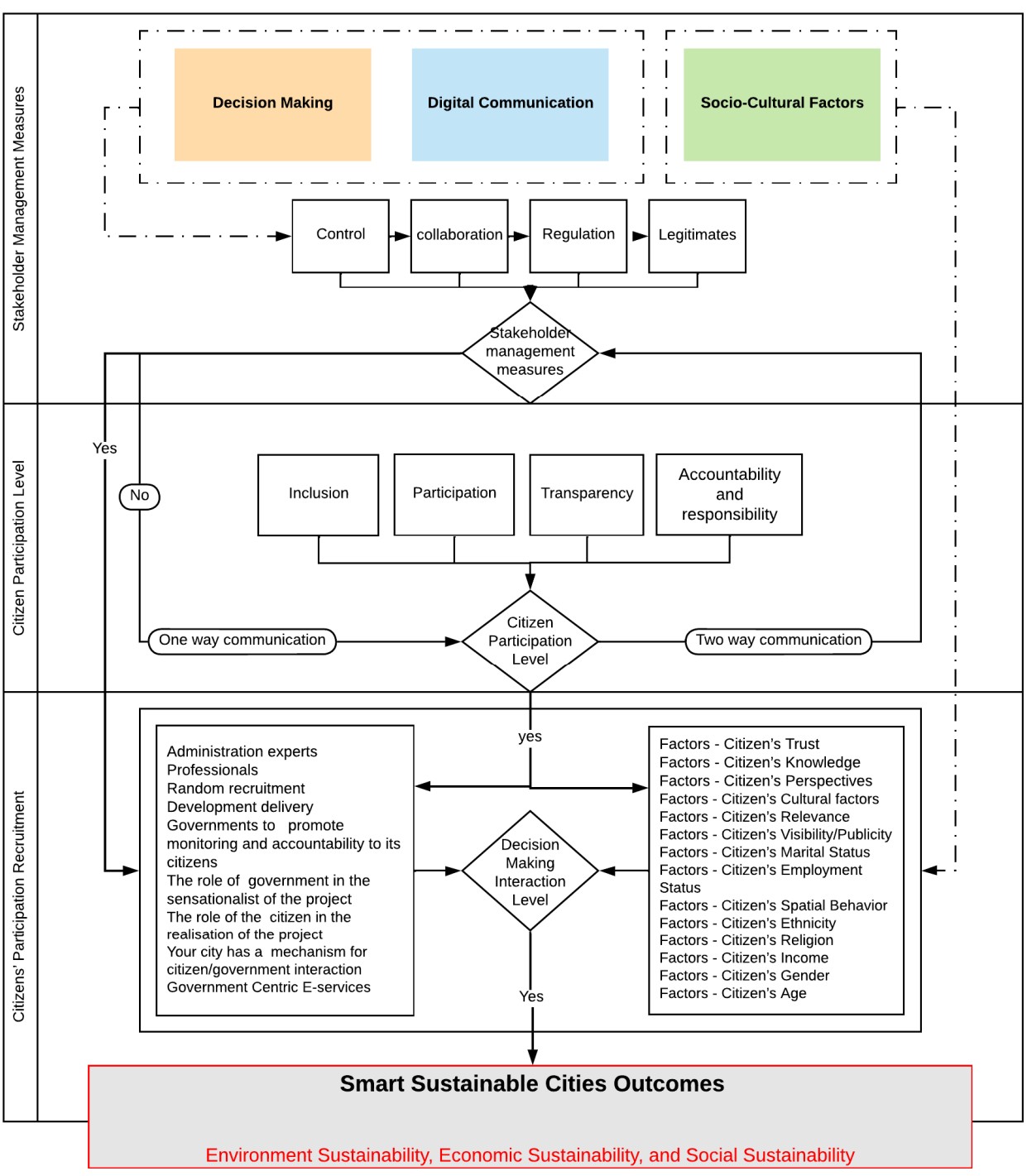

**Figure 2.** Citizen Participation for Smart Sustainable Cities Framework (CPSSCF).

On the other hand, CPL can help to ensure that the needs and perspectives of all members of the community are taken into account and that the resulting plan reflects the values and priorities of the community and drives SCCO. Four factors were considered in this framework [56]. Accountability and responsibility refer to the obligation of an individual or organisation to answer for their actions and decisions and to take ownership of their consequences. Transparency is the quality of being open and honest, providing access to information, and being easily understood. Participation refers to being actively engaged in a process, such as decision making or community engagement. Inclusion is the act of making sure that all individuals have an equal opportunity to participate and be heard, regardless of their background or identity. Together, these concepts promote good governance, democratic decision making, and social justice by ensuring that all voices are heard and all actions are accountable.

Lastly, CPR, an additional implementation measure identified by this study, is another important factor that sets the criteria to involve the most relevant participants. This recruitment can be achieved by involving two characteristics. First, the attributes of the participant are as follows: citizens' trust, knowledge, perspective, cultural factors, relevance, visibility publicity, marital factors, employment status, spatial behaviour, ethnicity, religion, income, gender, and age. Additionally, the quality of participants is crucial, such as in administration, random recruitment delivery, accountability, role of government, role of citizen in realisation of projects, and two-way interaction with government.

The expected outcomes are presented in Figure 2, which are Environmental Sustainability, Economic Sustainability, and Social Sustainability. It involves a holistic approach that addresses the interrelationship of these three areas. It also includes actions such as: developing policies and programs, implementing regulations and incentives, investing in the conservation and restoration of natural resources, fostering economic growth, advocating for social justice and protecting human rights, and encouraging community involvement and participation. It is a multi-stakeholder approach that involves government, business, and civil society working together to implement SSC solutions.

## 5. Conclusions

As determined through this study, SSC-related standards are being developed by various international scholars, organisations, and government entities within their specific domains, but little attention is given to citizens' participation. A framework for citizens' participation was developed with the objective of determining whether any additional stakeholder management measures are necessary, as well as explaining the relationship between SMM, CPL, and CPR. Three rounds of the Delphi method were conducted. The importance of these additional stakeholder management measures was confirmed by MS ranking and Kendall Coefficient. The results suggest three main component structures of the framework, which are SMM, CPL, and CPR. Essentially, CPR specifies the standards for recruiting suitable citizen participants, SMM motivates citizens to participate and raise CPL in the SCC development process, and all of these result in SSCO to achieve FSCP within the context of Vision2030 government policy.

The proposed framework is an essential contribution to understanding the issue conceptually and practically. This study is built upon theoretical foundations which can provide a solid foundation for understanding and addressing an issue of SSCO. Theoretical underpinnings can help explain why the Delphi method is being used. It can also provide a basis for evaluating the effectiveness of the framework. Developing a conceptual framework that can support the implementation of SSC will contribute to the body of knowledge by acknowledging the contribution of CP and understanding CPR standards.

The implication to knowledge is that the framework provides a structure for organizing and understanding existing knowledge, as well as for highlighting gaps in the current body of knowledge. This study proposes a framework where SMM is correlated to CPL, and CPL is also correlated to SSCO within the context of Saudi Arabia. Moreover, CPR is an essential foundation where it identifies the characteristics of participants. It can also be used as a

tool to guide and inform future research, helping to expand and deepen knowledge in the SSC field to cover other related aspects, as presented in Figure 1. Additionally, the developed framework can serve as a common language or reference point for researchers and practitioners in involving and facilitating communication and collaboration for CP in SSC.

Another implication is in providing a conceptual and practical framework for understanding the urban challenges. Policy implications include facilitating the decision makers' implementation of SSC in other cities and understanding stakeholder expectations. In other words, SSCO supports communication between the authorities and decision makers and identifies what citizens are expected to do. Moreover, this study recommends that authorities raise the level of CP by involving stakeholders' participation in smart cities by providing them with the opportunity to participate in the development of SSC regulations and engaging them in the process, as well as providing them with a basis for improving strategic management, which involves their participation at every stage. Subsequently, this research develops inclusive communication through the CPSSCF framework. Consequently, SSC will be developed and implemented.

For future research, the proposed framework should be explored for its scalability so that it can be aligned with VISION 2030 and government objectives and strategies as well as broader regulations to provide sight between government policies, citizens' requirements, and infrastructure asset performance in the future. To improve the overall generalisation, additional research should be conducted when the FSCP is fully implemented in 17 pilot cities. It is, therefore, important to be cautious when generalising the findings of this study. Moreover, the survey's respondents were limited to Saudi Arabians and a small study group, indicating that a more diverse sample is needed. The study was conducted with the aim of including female respondents, despite the majority of urban professionals in Saudi Arabia being male. Moreover, a solid connection between urban development and ordinary citizens could have been improved. Therefore, once women have a greater presence in the industry, future studies will explore their views on citizen participation. Furthermore, future research could include qualitative approaches such as focus groups and interviews for a more in-depth understanding. It is important to consider these limitations when examining how the CPSSCF framework could be adopted in developing countries in order for SSC to achieve success.

**Author Contributions:** Conceptualisation, A.K.A., R.B.A. and T.Y.M.L.; methodology, A.K.A., R.B.A. and T.Y.M.L.; software, A.K.A.; validation, A.K.A., R.B.A. and T.Y.M.L.; formal analysis, A.K.A.; investigation, R.B.A. and T.Y.M.L.; resources, A.K.A., R.B.A. and T.Y.M.L.; data curation, A.K.A.; writing—original draft preparation, A.K.A.; writing—review and editing, R.B.A. and T.Y.M.L.; visualisation, A.K.A., R.B.A. and T.Y.M.L.; supervision, R.B.A. and T.Y.M.L.; project administration, A.K.A., R.B.A. and T.Y.M.L. All authors have read and agreed to the published version of the manuscript.

**Funding:** This research received no external funding.

**Institutional Review Board Statement:** Not applicable.

**Informed Consent Statement:** Not applicable.

**Data Availability Statement:** It can be made available upon request to the corresponding author.

**Acknowledgments:** This paper forms part of a larger research project which focuses on citizen participation to support the implementation of smart sustainable cities, from which other papers will be produced with a different objective/scope but sharing the same background and methodology. The Saudi Arabia government, represented by Imam Abdulrahman Bin Faisal University (IAU), is appreciated for their internal financial sponsorship and other support for this PhD study.

**Conflicts of Interest:** The authors declare no conflict of interest.

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
