# Peer review of "Implementing Smart Sustainable Cities in Saudi Arabia: A Framework for Citizens’ Participation towards SAUDI VISION 2030"

_sustainability, doi:10.3390/su15086648_

Round 1
Reviewer 1 Report
The manuscript titled “Implementing Smart Sustainable Cities in Saudi Arabia: A Framework for Stakeholder Citizen’ Participation Towards SAUDI VISION 2030” has been reviewed entirely. This manuscript develop a citizens’ participation framework, identify any additional stakeholder’s management measures (SMM) (in addition to the ones previously developed by the authors), and explain the relationship with citizens’ participation level (CPL) for driving SSC. Three rounds of the Delphi method were conducted to structure and validate the results by experts in the field of built environment and reach a consensus of understanding the drivers of SSC. Mean score (MS) ranking and Kendall Coefficient were used to confirm the importance of these additional stakeholder’s management measures. The manuscript is interesting, however some parts of the manuscript needs some improvement before its acceptance for publication in this journal.
1. The abstract and conclusion should contain the main contribution of the paper clearly.
2. Some syntax errors need to be revised. The authors should check the manuscript carefully to avoid the typing and editing mistakes.
3- Improve the quality of figure1.
4- The conclusion is too long, you have to summarize it.
Author Response
Responses to comments from Reviewer 1.
|
S.No |
Reviewer |
Response |
|
1 |
The manuscript titled “Implementing Smart Sustainable Cities in Saudi Arabia: A Framework for Stakeholder Citizen’ Participation Towards SAUDI VISION 2030” has been reviewed entirely. This manuscript develop a citizens’ participation framework, identify any additional stakeholder’s management measures (SMM) (in addition to the ones previously developed by the authors), and explain the relationship with citizens’ participation level (CPL) for driving SSC. Three rounds of the Delphi method were conducted to structure and validate the results by experts in the field of built environment and reach a consensus of understanding the drivers of SSC. Mean score (MS) ranking and Kendall Coefficient were used to confirm the importance of these additional stakeholder’s management measures. The manuscript is interesting, however some parts of the manuscript needs some improvement before its acceptance for publication in this journal. |
Thank you very much for acknowledging the efforts of the authors. We do appreciate and respect the time the reviewer has spent in providing feedback to the authors to further help and improve this manuscript.
Based on the suggestions/comments, we have revised the manuscript accordingly. |
|
2 |
The abstract and conclusion should contain the main contribution of the paper clearly. |
Thanks for the suggestion. We have considered the main contribution of this study in the introduction and conclusion section. Please refer to page 1 lines 30-33 and page 17 lines 589 to 601.
|
|
2- |
Some syntax errors need to be revised. The authors should check the manuscript carefully to avoid the typing and editing mistakes. |
Thank you for pointing out this comment. We have carefully proofread the manuscript to avoid any syntax errors. |
|
3- |
Improve the quality of figure1.
|
The authors appreciate your feedback. The quality of the figure has been improved and enlarged. |
|
4- |
The conclusion is too long, you have to summarize it. |
Thanks for the suggestion. We have removed some part of the Conclusions section that is not needed in that section. |
Lastly, we would like to thank the reviewers for their thoughtful comments and efforts towards improving our manuscript. Much appreciated!
Reviewer 2 Report
In this article, the authors discuss challenges in creating smart, sustainable cities in Saudi Arabia. The authors provide a conceptual framework that enables SSC. This manuscript faces a challenge that forced me not to accept it for consideration in the Journal of Urban Design and Planning. Modifications can be influenced by the following points.
1- The title is well written outlining the scope of work and the expected outcome/s.
2- The introduction of minor changes must be carefully considered. The structure of your introduction should be revised. The literature lacks supporting references. I would recommend adding a paragraph after the first paragraph of the current introduction. This will discuss what scholars have written about the ongoing debate about smart sustainable cities and sustainable cities. We can see that the article refers to 'Smart Sustainable Cities'. A justification should also be provided.
Again, the introduction needs to redefine the research problem. Another paragraph that isn't well described in your introduction. The novelty of this study should also be considered. It is unclear whether the novelty lies in the framework of smart, sustainable cities or in the methods used to conduct such frameworks.
3- The literature review: The methods used to review this are unclear. The authors should describe their research questions prior to conducting data collection. The research should also explain in which database the framework is built and what the inclusion and exclusion criteria are. The authors should describe what has been done.
4- The research methodology should specify how many rounds of the Delphi technique were used to obtain the data from stackholders. Sometimes I understand there are two rounds and sometimes I recognize there are more than three rounds. Describing the rounds of this technique in its first mention in the methodology is extremely significant. In addition, it is very critical to mention this issue in the introduction clearly and briefly. It is also unclear how the snowballing technique was used with Dephi. Did it appear in the first round or in all rounds? How did the authors of this study deal with the same sample size if it was applied to all rounds? What was the purpose of each round?
It is advisable to restructure the methodology section to provide a clearer understanding of how the Delphi was carried out. This should be followed by a description of the methodology and then an explanation of how each round of Delphi was conducted. In this context I would recommend deleting the text from 255-288 on Page 8.
5- The results were well explained and very helpful in explaining the main findings. However, the results and discussion section failed to link findings with prior studies that use similar methods to address the same challenges in cities. Here I would recommend the following articles that could help:
DOI: 10.1016/j.scs.2021.102782
DOI: 10.1108/IJBPA-07-2018-0055
Reference [54] of Figure 1 needs to be revised. This digarme does not belong to this reference. Please double check.
6- A typo that need further correction:
- "For example. in 2016 the Future Saudi Cities Program (FSCP)..." which should has a comma after 'For example'
- "In addition, Cities can boost their.." which should be "In addition, cities can boost their.."
- "fourth section results discussions and validation ..." missing commas are in this sentenece.
- "... in order to achieve SSCO" a missing fullstop.
Author Response
Responses to comments from Reviewer 2.
|
S.No |
Reviewer |
Response |
|
1 |
In this article, the authors discuss challenges in creating smart, sustainable cities in Saudi Arabia. The authors provide a conceptual framework that enables SSC. This manuscript faces a challenge that forced me not to accept it for consideration in the Journal of Urban Design and Planning. Modifications can be influenced by the following points. |
We do appreciate and respect the time the reviewer has spent in providing feedback to the authors to further help and improve this study. We made all the possible changes to improve the manuscript. |
|
1 |
The title is well written outlining the scope of work and the expected outcome/s. |
Thank you very much for acknowledging the efforts of the authors. |
|
2.1 |
The introduction of minor changes must be carefully considered. The structure of your introduction should be revised. The literature lacks supporting references. I would recommend adding a paragraph after the first paragraph of the current introduction. This will discuss what scholars have written about the ongoing debate about smart sustainable cities and sustainable cities. We can see that the article refers to 'Smart Sustainable Cities'. A justification should also be provided. |
The introduction has been improved by adding a paragraph after the first paragraph. This paragraph discusses the debate about 'Smart Sustainable Cities” and it was supported by references and justifications. Please refer to page 2 lines 49 to 60. |
|
2.2 |
Again, the introduction needs to redefine the research problem. Another paragraph that isn't well described in your introduction. The novelty of this study should also be considered. It is unclear whether the novelty lies in the framework of smart, sustainable cities or in the methods used to conduct such frameworks. |
Thank you for your feedback. We have redefined the research problem. It can be found on page 3 lines 132 to 137.
In addition, we have reemphasized the novelty of this study. It can be found on page 4 lines 157-167. |
|
3- |
The literature review: The methods used to review this are unclear. The authors should describe their research questions prior to conducting data collection. The research should also explain in which database the framework is built and what the inclusion and exclusion criteria are. The authors should describe what has been done.
|
The author would like to emphasise that the research question is described in page 4 line 152-153. From that question we developed the hypothesis which is shown on page 7 lines 256 to 259 and supported by the literature review with previous studies and framework. Also, it presents the limitation of each framework and why our study is needed to develop a novel framework. However, the criteria for inclusion and exclusion was not mentioned in this study due to words limitation. We were granted ethics approval from the University of New South Wales, Sydney, and such information is presented and approved in that ethics. The authors are happy to provide any additional information the reviewer or the editors may need. That being said, a short explanation has been added on page 4 lines 157-167 |
|
4.1 |
The research methodology should specify how many rounds of the Delphi technique were used to obtain the data from stackholders. Sometimes I understand there are two rounds and sometimes I recognize there are more than three rounds. Describing the rounds of this technique in its first mention in the methodology is extremely significant. In addition, it is very critical to mention this issue in the introduction clearly and briefly. It is also unclear how the snowballing technique was used with Delphi. Did it appear in the first round or in all rounds? How did the authors of this study deal with the same sample size if it was applied to all rounds? What was the purpose of each round?
|
Thank you for your valuable comments. We have considered them accordingly as follow: 1- The adopted methodology is now mentioned clearly and briefly in the introduction Page 4 line 152-153. 2- This study follow three round of Delphi to fulfill the research aim and question. 3- Participant were selected based on judgmental sampling technique. However, the snowball technique was proposed in case any participant withdrawal during the data collection of the three rounds. If one participant withdraws, they will be gently asked to propose another participate. 4- The purpose of this study is to develop a framework by reaching a consensus among experts. Therefore, it is mandatory that the same participant complete the three rounds. Fortunately, no withdrawal appeared in this study. 5- The following is the purpose of each round of Delphi method as mentioned on pages 8-9: lines 270-282: A- First Round: confirm the framework and identify any additional measures. Line 278. B- Second Round: validation of the responses of the structure of the framework by comparing it with the consolidated of others’ opinion. Line 288. C- Third Round: participants were asked an open-ended question to confirm and identify the influencing measures and to explain the relationships. Lines 292-293 |
|
4.2 |
It is advisable to restructure the methodology section to provide a clearer understanding of how the Delphi was carried out. This should be followed by a description of the methodology and then an explanation of how each round of Delphi was conducted. In this context I would recommend deleting the text from 255-88 on Page 8. |
Thank you for your advice. The methodology section has been restructured and revised accordingly. |
|
5 |
The results were well explained and very helpful in explaining the main findings. However, the results and discussion section failed to link findings with prior studies that use similar methods to address the same challenges in cities. Here I would recommend the following articles that could help: DOI: 10.1016/j.scs.2021.102782 DOI: 10.1108/IJBPA-07-2018-0055 Reference [54] of Figure 1 needs to be revised. This digarme does not belong to this reference. Please double check. |
Thank you for the recommendation of the insightful articles.
The quality of the Figure 1 has been improved and enlarged. |
|
6 |
A typo that need further correction: - "For example. in 2016 the Future Saudi Cities Program (FSCP)..." which should has a comma after 'For example' - "In addition, Cities can boost their.." which should be "In addition, cities can boost their.." - "fourth section results discussions and validation ..." missing commas are in this sentenece. - "... in order to achieve SSCO" a missing fullstop. |
Thank you for pointing this out, typos have been revised accordingly. |
Lastly, we would like to thank the reviewers for their thoughtful comments and efforts towards improving our manuscript. Much appreciated!
Reviewer 3 Report
This is an impressive undertaking and the authors do a very good job of explicating and linking the conceptual models to create a holistic model of citizen participation in smart sustainable cities.
My only concern is with the way that the Delphi method is used here to determine to composite model. Although I think that the Delphi method itself is not necessarily a problem, the N of 25 is simply too small to lend the results much credibility. Either a larger N or better yet, several Delphi groups, each with the three rounds that the authors use, would make for more compelling results. Particularly since the authors were not able to get a diverse sample (with respect to gender) with the first group chosen, more groups, including groups chosen specifically to achieve gender diversity, would bolster the case.
Author Response
Responses to comments from Reviewer 3.
|
S.No |
Reviewer |
Response |
|
1 |
This is an impressive undertaking and the authors do a very good job of explicating and linking the conceptual models to create a holistic model of citizen participation in smart sustainable cities. |
Thank you very much for acknowledging the efforts of the authors. We do appreciate and respect the time the reviewer has spent in providing feedback to the authors to further help and improve this manuscript.
|
|
1 |
My only concern is with the way that the Delphi method is used here to determine to composite model. Although I think that the Delphi method itself is not necessarily a problem, the N of 25 is simply too small to lend the results much credibility. Either a larger N or better yet, several Delphi groups, each with the three rounds that the authors use, would make for more compelling results. Particularly since the authors were not able to get a diverse sample (with respect to gender) with the first group chosen, more groups, including groups chosen specifically to achieve gender diversity, would bolster the case. |
Thank you for your feedback in terms of the sample size and the gender diversity.
First, in term of sample size, this study focuses on the perspective of decision-makers toward CP. The purpose of data collected from the Delphi method is to develop and validate the framework by decision making in the field of built environment and reach a consensus among stakeholders. Please refer to page 18 lines 628 to 629. Therefore, the participants were limited to 25. According to Okoli and Pawlowski (2004), Delphi method does not depend on statistical power, but rather on dynamics for arriving at consensus among experts. Therefore, it is recommended to survey 10 and18 experts. Akins et al. (2005) argued that a small sample of experts who have effective and reliable utilisation of the field would lead to rigorous judgment and effective decision making.
Second, in term of gender diversity, the authors acknowledge that equitable and egalitarian must exist in any scientific study. We have appropriately justified that in the manuscript on page 9 line 316-322. However, the lack of participant diversity is due to the social setting of Saudi Arabia. It is hoped that in future women participation in such project and surveys will be improved. Although most urban professions in Saudi Arabia are dominated by males, we have done our best to capture both genders. According to The World Bank (2022), Saudi Arabia has just started to embrace and empowering women in the executive positions. Nevertheless, these limitations will be addressed in future research when women’s representation in the industry improves. Please see page 19 line (632-634). 1- Abdelmoaty, A., Saadallah, D., & Bakr, A. (2021). Gender Mainstreaming and Women’s Involvement in Urban Planning Strategies. 2- Al-Hazzaa, H. M. (2018). Physical inactivity in Saudi Arabia revisited: A systematic review of inactivity prevalence and perceived barriers to active living. International journal of health sciences, 12(6), 50. 3- Naseem, S., & Dhruva, K. (2017). Issues and challenges of Saudi female labor force and the role of Vision 2030. International Journal of Economics and Financial Issues, 7(4), 23-27. That being said, this limitation will be taken into consideration in future studies |
The World Bank. Labor force, female (% of total labor force) - Saudi Arabia. Available online: https://data.worldbank.org/indicator/SL.TLF.TOTL.FE.ZS?locations=SA (accessed on 22/11/2022).
Okoli, C., & Pawlowski, S. D. (2004). The Delphi Method as a Research Tool: An Example, Design Considerations and Applications. Information & Management, 42(1), 15-29.
Akins, R., et al. (2005). Stability of response characteristics of a delphi panel: application of bootstrap data expansion. BMC Medical Research Methodology, 5, 37. doi:10.1186/1471-2288-5-37
Lastly, we would like to thank the reviewers for their thoughtful comments and efforts towards improving our manuscript. Much appreciated!
Round 2
Reviewer 2 Report
After reading this revised version, I see the authors have provided proper responses and made development changes. The authors have obviously taken the feedback they received into consideration and made sure to address any issues that were raised. They have provided more evidence and additional explanations to support their claims, and have made sure to update any outdated information. Thank you
Author Response
|
S.No |
Reviewer |
Response |
|
1 |
After reading this revised version, I see the authors have provided proper responses and made development changes. The authors have obviously taken the feedback they received into consideration and made sure to address any issues that were raised. They have provided more evidence and additional explanations to support their claims, and have made sure to update any outdated information. Thank you
|
Thank you very much for acknowledging the efforts of the authors. We do appreciate and respect the time the reviewer has spent in providing feedback to the authors to further help and improve this manuscript.
|
Lastly, we would like to thank the reviewers for their thoughtful comments and efforts towards improving our manuscript. Much appreciated!
Reviewer 3 Report
My concern with the initial submission was that the study group is a tiny, homogenous group. I suggested expanding / repeating the Delphi process with other groups. The authors did not do this; perhaps that would have required too much effort.
It certainly is interesting work and others could perhaps use the method in other settings. If this caveat is stated at the outset of the piece, I would have no trouble accepting it for publication.
Author Response
|
S.No |
Reviewer |
Response |
|
1 |
My concern with the initial submission was that the study group is a tiny, homogenous group. I suggested expanding / repeating the Delphi process with other groups. The authors did not do this; perhaps that would have required too much effort. It certainly is interesting work and others could perhaps use the method in other settings. If this caveat is stated at the outset of the piece, I would have no trouble accepting it for publication.
|
Thank you very much for this valuable comment. We would like to emphasize that the study group size and the homogeneous group are considered as a limitation that has been mentioned in section 3.1 page 9 lines 308- 309. Also, It was mentioned again in the conclusions section (5), and it can be found on page 18 lines 620-621. In addition, we have stated it directly at the outset of the piece where it can be found on page 1, lines 23-24
That being said, this limitation will be taken into consideration in future studies
|
Lastly, we would like to thank the reviewers for their thoughtful comments and efforts towards improving our manuscript. Much appreciated!